# Donkey Oil-Based Ketogenic Diet Prevents Tumor Progression by Regulating Intratumor Inflammation, Metastasis and Angiogenesis in CT26 Tumor-Bearing Mice

**DOI:** 10.3390/genes14051024

**Published:** 2023-04-30

**Authors:** Huachen Zhang, Lan Xie, Ning Zhang, Xingzhen Qi, Ting Lu, Jingya Xing, Muhammad Faheem Akhtar, Lanjie Li, Guiqin Liu

**Affiliations:** 1College of Agronomy, Shandong Engineering Technology Research Center for Efficient Breeding and Ecological Feeding of Black Donkey, Shandong Donkey Industry Technology Collaborative Innovation Center, Liaocheng University, Liaocheng 252000, China; 2Biopharmaceutical Research Institute, Liaocheng University, Liaocheng 252000, China; 3Inner Mongolia Key Laboratory of Equine Genetics, Breeding and Reproduction, Equine Research Center, College of Animal Science, Inner Mongolia Agricultural University, Hohhot 010018, China; 4Office of International Programs, Liaocheng University, Liaocheng 252000, China

**Keywords:** donkey oil, ketogenic diet, CT26^+^, colon cancer, HIF-1*α*

## Abstract

Colon cancer is one of the typical malignant tumors, and its prevalence has increased yearly. The ketogenic diet (KD) is a low-carbohydrate and high-fat dietary regimen that inhibits tumor growth. Donkey oil (DO) is a product with a high nutrient content and a high bioavailability of unsaturated fatty acids. Current research investigated the impact of the DO-based KD (DOKD) on CT26 colon cancer in vivo. Our findings revealed that DOKD administration significantly lowered CT26^+^ tumor cell growth in mice, and the blood *β*-hydroxybutyrate levels in the DOKD group was significantly higher than those in the natural diet group. Western blot results showed that DOKD significantly down-regulated Src, hypoxia inducible factor-1*α* (HIF-1*α*), extracellular signal-related kinases 1 and 2 (Erk1/2), snail, neural cadherin (N-cadherin), vimentin, matrix metallopeptidase 9 (MMP9), signal transducer and activator of transcription 3 (STAT3), and vascular endothelial growth factor A (VEGFA), and it significantly up-regulated the expressions of Sirt3, S100a9, interleukin (IL)-17, nuclear factor-kappaB (NF-*κ*B) p65, Toll-like receptor 4 (TLR4), MyD88, and tumor necrosis factor-α. Meanwhile, in vitro validation results showed that LW6 (a HIF-1*α* inhibitor) significantly down-regulated the expressions of HIF-1*α*, N-cadherin, vimentin, MMP9, and VEGFA, which supported those of the in vivo findings. Furthermore, we found that DOKD inhibited CT26^+^ tumor cell growth by regulating inflammation, metastasis, and angiogenesis by activating the IL-17/TLR4/NF-*κ*B p65 pathway and inhibiting the activation of the Src/HIF-1*α*/Erk1/2/Snail/N-cadherin/Vimentin/MMP9 and Erk1/2/HIF-1*α*/STAT3/VEGFA pathways. Our findings suggest that DOKD may suppress colon cancer progression and help prevent colon cancer cachexia.

## 1. Introduction

Colon cancer is a typical digestive tract malignancy with an increasing incidence year by year [1]. Treatment strategies in colorectal cancer include surgery, chemotherapy, and radiation [2], but these therapies have many limitations, adverse effects and a high recurrence rate. Therefore, searching for a new strategy to treat colon cancer is necessary. The metabolic pattern of aerobic glycolysis in cancer cells can be a potent target for cancer therapy, and much of the research on anti-tumor therapies revolves around tumor metabolism such as anti-angiogenic treatment [3]. The ketogenic diet (KD) has recently been investigated as an adjuvant cancer therapy, which is a diet that is enriched in fat, low in carbohydrates, and moderate in protein and other nutrients [4]. KD has been used to limit glucose metabolism and thus inhibit cancer progression. Moreover, KD slows tumor advancement by enhancing immune defense; inhibiting cell proliferation, cell metastasis, and angiogenesis; and promoting tumor apoptosis [5]. Notably, a deluge of literature has enunciated the anti-tumor activity of KD against glioma [6], colon cancer [7], prostate cancer [8], and endometrial cancer [9]. It has been found in the studies of various KD formulas for the adjuvant treatment of colon cancer that KD can affect glucose metabolism to delay tumor angiogenesis, hinder tumor proliferation and metastasis, and delay the growth of colon cancer [7,10]. This kind of diet therapy has become a hot topic in cancer therapy due to its low side effects and low cost.

Appropriate carbohydrate restriction and ketogenic ratio and the duration of the KD are different for patients [11,12], sometimes, KD is difficult to implement. Therefore, it is necessary to improve the KD formula. Donkey oil (DO) is consumed as edible oil, and it has considerable importance in medicine and health care. Compared with lard, tallow, and goat, donkey oil has high levels of unsaturated fatty acids and essential fatty acids, such as oleic acid (32.30%), linoleic acid (12.90%), and palmitic acid (26.33%), along with a low content of saturated fatty acids. It has certain nutritional advantages [13,14]. Moreover, donkey oil has a high concentration of vitamin E, which is essential for the immune system and metabolism [15]. It is characterized by low cholesterol, which may benefit patients with cardiovascular disease and some types of cancer [16]. Based on the characteristic of donkey oil, we speculated that the application of donkey oil in KD could elevate the anti-tumor effect of KD. Currently, the impact of DO-based KD (DOKD) on cancer has not previously been reported.

The KD’s inhibitory effect on tumors is mainly achieved by affecting metabolism. The KD uses ketone bodies instead of glucose to supply energy; changes the homeostasis of ketone bodies by regulating glucose transport 1, insulin-like growth factor-1 and other levels [17]; and regulates related pathways to inhibit tumor progression in aspects of inflammation, oxidative stress, metastasis and angiogenesis. Inflammation was originally thought to be a host response to the tumor, leading to tumor suppression. However, chronic inflammation is associated with poor clinical outcomes. Pro-inflammatory cytokines, such as interleukin (IL)-1*β* and IL-6, are highly expressed in cancer patients and are thought to promote an immunosuppressive tumor microenvironment [18]. Inflammatory cytokines, such as tumor necrosis factor-α (TNF-*α*), and T cells secrete cytokines such as interferon-*γ* (IFN-*γ*), IL-2, IL-17, IL-22, and IL-36, all produced by macrophages. These cells/cytokines exert their antitumor affect by enhancing an immune response [19,20]. Some data suggest that inflammation and hypoxia in the tumor microenvironment are essential components for tumor progression and the metastatic cascade [21]. Hypoxia inducible factor-1α (HIF-1α) is the critical regulator of the hypoxia response and is involved in one of the most common oncogenic pathways, the PI3K/AKT/mTOR pathway, and is associated with immune responses, inflammation, angiogenesis, and metastasis. It is considered a target for cancer therapy [22,23]. Nevertheless, regulations regarding HIF-1*α* with DOKD in cancer activity have yet to be elucidated.

Metastasis, a hallmark of cancer, is the primary reason for mortality in cancer patients [24]. Neural cadherin (N-cadherin) is one of the classic cadherins associated with tumor progression, increased metastasis, and the invasive behavior of cancer cells [25,26]. In addition, vimentin is a widely expressed protein of the type IIIIF protein family, which increases expression in multiple tumor cell lines and tissues [27]. N-cadherin and vimentin are essential in promoting cancer and mesenchymal transformation (EMT) induction [27,28]. EMT is involved in numerous pathological changes, including tumor cell invasion and metastasis [29,30]. HIF-1α has been reported to regulate the expression of various EMT markers and modulators, such as transcriptional activation of N-cadherin and snail. In addition, matrix metalloproteinase 9 (MMP9) plays an essential role in extracellular matrix remodeling, which is critical in cell migration, invasion, and angiogenesis, promoting tumor progression [31,32,33]. At hypoxic tumor sites, HIF-1*α* activates MMP-2 and -9 to promote tumor metastasis. Therefore, targeting HIF-1*α* has now become the main focus of drug development for cancer treatment. Tumor angiogenesis is vital in tumor growth metastasis, transporting nutrients and oxygen to cancer cells [34]. Vascular endothelial growth factor A (VEGFA), a leading simulator of tumor-initiated angiogenesis, has been found to be overexpressed in many cancers [35], such as breast cancer [36], lung cancer [37], and colon cancer [38]. It has been reported that HIF-1*α* can activate VEGFA and promote tumor angiogenesis. Moreover, signal transduction and transcription activator 3 (STAT3) induces the expression of factors that promote angiogenesis, invasiveness, and metastasis, such as MMPs and VEGFA [39,40]. STAT3 has also been reported to induce the expression of HIF-1*α* and cause tumor angiogenesis.

In this study, we made DOKD for the first time. To determine the effect of DOKD on CT26^+^ colon cancer, we compared its effects with a natural diet (ND) on the growth of a mouse model of CT26^+^ colon cancer, and demonstrated the anti-tumor effect of DOKD on colon cancer in CT26^+^ mice and explored its potential anti-tumor mechanisms by determining inflammation, migration, and angiogenesis. These results preliminarily revealed the antitumor mechanism of DOKD. In addition, we also examined the effect of HIF-1*α* on anti-tumor metastasis and angiogenesis pathways at the cellular level.

## 2. Materials and Methods

### 2.1. Cell Line and Culturing

Mouse colon cancer (CT26^+^) cell line was bought from RIKEN Biological Resource Center, Tsukuba, Japan. In this experiment, RPMI1640 medium containing L-glutamine, 10% fetal bovine serum and 100 units/mL penicillin-streptomycin (all purchased from Thermo Fisher, Waltham, MA, USA) was used to culture cells. The culture conditions were 37 °C and 5% CO_2_.

### 2.2. Mice and Tumor Implantation

A total of 12 BALB/c male mice aged 7 weeks with a body weight of 21 g ± 0.1 g, bought from Jinan Pengyue Experimental Animal Company (Jinan, China), were used in the present study. About one million CT26^+^ cells were subcutaneously inoculated into the BALB/c mice with a 27 G needle. All mice had free access to food and water and were kept in a dark cycle of 12 h/12 h. All animal procedures were carried out by the Guidelines for Care and Use of Laboratory Animals of Liaocheng University, and the experimental protocol was reviewed and approved by the Animal Care and Use Committee of Liaocheng University (permit number 2021111030).

### 2.3. DOKD Treatments

Experimental treatments were stated on the 14th day after the CT26^+^ cell injection. After acclimation, 14 mice were divided into 2 groups: ND (n = 6) and DOKD (n = 6). The ND groups used the standard rodent diet, AIN-93G, and the DOKD group used the DOKD ad libitum for ten days. The compositions of the diets are listed in Table 1.

### 2.4. Measurement of Body Weight and Tumor Volume

During the whole animal experiment, we measured the body weight of every mouse daily, and we measured the tumor volume of every mouse every two days. Following this, the tumor tissue of each mouse was taken and weighed. The weight and volume of tumors were measured, and the tumor tissues were harvested and frozen in liquid nitrogen.

### 2.5. Measurement of β-Hydroxybutyrate and Glucose Levels

Blood serum samples were used to determine *β*-hydroxybutyrate and glucose concentrations via handheld blood glucose and blood ketone meter (JPST-5 blood glucose and blood ketone meter, Beijing Yicheng bioelectronics Co., Ltd. Beijing, China).

### 2.6. Transcriptome Sequencing (RNA-Seq)

RNA-seq was performed on six tumor samples by Shandong Jiehelix Biotechnology Co., LTD. Transcriptome sequencing included two groups of treatments, each with three replicates, and a total of six cDNA libraries were constructed. RNA sample purity was detected by NanoPhotometer spectrophotometer (Thermo Fisher, Waltham, MA, USA) and Agilent 2100 RNA Nano 6000 Assay Kit (Agilent Technologies, Santa Clara, CA, USA) to determine the integrity and concentration of RNA samples. After confirming that the quality of the total RNA sample was qualified, the Illumina HiSeqTM4000 high-throughput sequencing platform (Thermo Fisher, Waltham, MA, USA) was used to transcriptome sequence the established library using PE150. After filtering the raw data (raw reads) off the machine, high-quality clean reads were obtained after removing low-quality sequences to remove joint contamination, etc. Trinity software was used to assemble and splice clean reads to obtain unigene [41]. Then, we performed the gene ontology (GO) enrichment analysis and Kyoto Encyclopedia of Genes and Genomes (KEGG) pathway analysis. RSEM (http://deweylab.github.io/RSEM/, accessed on November 15, 2022) was used to calculate the level of gene expression of the transcriptome with reuse DESeq2 analysis software (Anders and Huber, 2010), and |log2 Fold Change| ≥ 1 with *p* < 0.05 was the differentially expressed mRNA among the selected samples. BLAST software compared the differentially expressed gene sequences with Swiss-Prot, Nr, COG, GO, KOG, KEGG, and eggNOG4 databases (E value < 1 × 10^−5^) to obtain the corresponding annotation information. Then, the software GOseqGO was used to complete the enrichment analysis [42]. In the enrichment analysis of the KEGG pathway completed by Cluster Profiler, *p* < 0.05 indicated a significantly enriched KEGG pathway.

### 2.7. Western Blot (WB)

According to the manufacturer’s instructions, the total tumor proteins were extracted using the Minute total protein extraction kit (Invent Biotechnologies) combined with 100× protease inhibitor cocktails (CW2200S; Beijing ComWin Biotech Co., Ltd., Beijing, China) and 100× phosphatase inhibitor cocktail (CW2383S; Beijing ComWin Biotech Co., Ltd., Beijing, China). The BCA protein assay kit (CW0014S; Beijing ComWin Biotech Co., Ltd., Beijing, China) was used to determine tumor protein concentrations. Bio-Rad Mini-Protein II system was used to separate the same amount of protein (25 mg/lane) by sodium dodecyl sulphate-polyacrylamide gel electrophoresis (SDS-PAGE) [43]. The proteins were isolated by SDS-PAGE and then transferred to polyvinylidene fluoride membranes (Millipore, Billerica, Burlington, MA, USA). The membrane was incubated with 5% skim milk in a tris-buffered room temperature for 2 h and incubate with primary antibody at 4 °C overnight. The following antibodies were used: HIF-1*α* (ab179483, 1:1000, Abcam), Sirt3 (WL03840, 1:1000, Wanleibio), MMP-9 (ab228402, 1:1000, Abcam), p-Src (WL000096, 1:1000, Wanleibio), Src (ab185617, 1:5000, Abcam), Erk1/2 (ab17942, 1:1000, Abcam), p-Stat3 (ab267373, 1:1000, Abcam), STAT3 (10253-2-AP, Proteintech, 1:2000), Snail (ab180714, 1:1000, Abcam), N-Cadherin (ab76011, 1:10,000, Abcam), Vimentin (ab92547, 1:5000, Abcam), VEGFA (ab46154, 1:1000, Abcam), NF-*κ*B p65 (66535-lg, 1:1000, Proteintech), MYD88 (ab219413, 1:1000, Abcam), S100a9 (ab242945, 1:1000, Abcam), IL-17 (ab79056, 1:1000, Abcam), TLR4 (66350-1-Ig, 1:5000, Proteintech), TNF-*α* (17590-1-AP, 1:1000, Proteintech) and *β*-actin (66009-1-Ig, 1:10,000, Proteintech). On the second day, these membranes were rinsed three times and then incubated with anti-rabbit IgG (H + L) secondary antibody (1:10,000, Proteintech) and anti-mouse IgG (H + L) secondary antibody (1: 10,000, Proteintech) at room temperature for 1.5 h. Immunoreactive proteins were observed by chemiluminescence ECL Western blot assay (Amersham Biosciences, Piscataway, NJ, USA). The strip density was represented by scanning units, and the expression level was quantified by comparison with the control level.

### 2.8. Cell Counting Kit-8 (CCK8) Assay

LW6 (a HIF-1*α* inhibitor) was purchased from MedChemExpress (Monmouth Junction, NJ, USA). According to the manufacturer’s protocols, cell viability was assessed by CCK8 (HY-K0301, MedChemExpress, Monmouth Junction, NJ, USA) assay [44]. The CT26^+^ cells were seeded in 96-well microplates at a density of 5 × 10^3^ per well. Cells were treated with different concentrations of LW6. After 24 h, 10 μL of CCK8 reagent was added to all wells, and incubation was continued for two hours. Three replicate wells were set up for each experiment. Using a microplate reader, we analyzed the absorbance of each well at 450 nm. Wells without cells were used as blanks.

### 2.9. Statistical Analysis

SPSS 20.0 software (https://www.ibm.com/support/pages/how-cite-ibm-spss- statistics-or-earlier-versions-spss, accessed on December 19, 2022) was used for statistical analysis, and the independent sample *t*-test was used to statistical analyze group differences. Data were expressed as mean ± mean standard error (SEM). Statistical analysis and mapping were performed using GraphPad Prism software (GraphPad software Inc., San Diego, CA, USA). Statistical significance: * *p* < 0.05; ** *p* < 0.01; *** *p* < 0.001.

## 3. Results

### 3.1. KD Inhibits CT26^+^ Colon Tumor Growth

BALB/c mice were decapitated on day 24 (Figure 1a). Tumor weights and volumes were sharply reduced in the DOKD group compared with those in the ND group (Figure 1c–e, ** *p* < 0.01), suggesting that DOKD significantly inhibits tumor development in vivo. During the 10 days from day 14 to day 24, there was no significant difference in the final body weight between the two groups with ND/DOKD treatments (Figure 1b).

### 3.2. Effects of DOKD on Blood β-Hydroxybutyrate and Blood Glucose in CT26^+^ Bearing Mice

Peripheral blood was collected for blood *β*-hydroxybutyrate and blood glucose analysis. Our results showed that the blood *β*-hydroxybutyrate was significantly higher in the DOKD group (Figure 2a, * *p* < 0.05 and ** *p* < 0.01). There were no significant differences between the DOKD/ND groups (Figure 2b). It was suggested that DOKD effectively increased blood *β*-hydroxybutyrate levels, and DOKD did not affect blood glucose.

### 3.3. Differentially Expressed Genes (DEGs) of the DOKD/ND Group

We next examined the differentially expressed genes by GO enrichment analysis. In this study, eighteen DEGs (six down-regulated, including Gzmg, Gzmf, Gzmd, Gzme, and Prdm1, and twelve up-regulated, such as S100a8, S100a9, G0s2, Cxcl 2, and Cxcl 3 in DOKD mice) were screened out by analyzing the RNA-seq data (Figure 3). Transcripts with|log2 Fold Change|≥1 and *p* < 0.05 were screened as DEGs.

### 3.4. GO and KEGG Analysis

Based on the GO annotation, 18 genes were assigned GO numbers. Through GO analysis, a total of 170 GO items with *p* < 0.05 were received, including two cell component entries, 124 biological process entries, and 44 molecular function entries. In the cell components, the DEGs regulate the extracellular space, extracellular region, and nucleoplasm. In biological processes, the DEGs regulate the apoptotic process, immune response, neutrophil activation, etc. In molecular functions, the DEGs regulate the cytokine activity, toll-like receptor binding, etc. To better analyze the gene ontology enrichment of these 18 DEGs, the top 34 items with the most significant *p*-value for each component were shown in Figure 4a.

KEGG pathway analysis showed that DEGs were enriched in 18 signaling pathways, such as “The IL-17 signaling pathway”, “TNF signaling pathway”, “Cytokine-cytokine receptor interaction”, “Chemokine signaling pathway”, and other vital pathways. The top nine pathways with a significant *p*-value were shown in a bubble chart (Figure 4b).

### 3.5. Therapeutic Pathways and Potential Targets of DOKD on Mouse CT26^+^ Colon Cancer

Based on the recent research on the nosogenesis of CT26^+^-bearing mice with ND/DOKD and the results of the KEGG pathway analysis, we further structured the underlying IL-17, toll-like receptor 4 (TLR4), HIF-1α, nuclear factor-kappaB (NF-κB) p65 and TNF signaling as the therapeutic pathways in the DOKD treatment of CT26-bearing mice (Figure 5). This network shows significant underlying signaling pathways in the DOKD treatment of CT26^+^-bearing mice. Notably, it provides evidence revealing that DOKD may play a beneficial role by improving inflammation, immune response, metastasis and angiogenesis in CT26^+^-bearing mice.

Our data from KEGG showed that five signaling pathways, including IL-17, HIF-1*α*, TLR4, NF-*κ*B p65, and TNF signaling, may be the therapeutic pathways in the DOKD treatment of tumor development. Moreover, these five signaling pathways are closely associated with inflammation response, tumor metastasis, and angiogenesis [45,46,47,48]. Thus, we further investigated whether the tumor inflammation/immune response, metastasis, and angiogenesis were influenced by KD treatment.

### 3.6. DOKD Up-Regulate the Expressions of IL-17, TLR4, NF-κB p65 and TNF-α and Down-Regulate the Expressions of HIF-1α in CT26^+^ Bearing Mice

To determine if tumor inflammation was influenced by IL-17, HIF-1*α*, and NF-*κ*B p65 expression in response to DOKD treatment, WB was performed. Our data showed that the terms of S100a9, IL-17, NF-*κ*B p65, MYD88, TLR4, and TNF-α in the DOKD mice group were significantly increased compared to that of the ND mice group (Figure 6b–h; * *p* < 0.05 and ** *p* < 0.01), while the expression level of HIF-1*α* was significantly decreased (Figure 6d, *** *p* < 0.001). This was consistent with our RNA-seq results. Our results suggested that KD enhances tumor inflammation partly due to IL-17, TLR4, and NF-*κ*B p65 activation.

### 3.7. DOKD Inhibits the Expression of Metastasis-Related Proteins in CT26^+^ Bearing Mice

In this study, we found that p-Src, extracellular signal-related kinases 1 and 2 (Erk1/2), snail, N-cadherin, MMP9 and vimentin expressions were significantly decreased in the DOKD group compared with the ND group (Figure 7b–i, * *p* < 0.05, ** *p* < 0.01 and *** *p* < 0.001), and the expression level of Sirt3 was significantly increased (Figure 7e, * *p* < 0.05). These results suggested that DOKD delayed the metastasis and invasion of CT26^+^ colon cancer cells partly via the Src/HIF-1*α*/Erk1/2/Snail/MMP9 signaling pathway in vivo.

### 3.8. DOKD Inhibits Tumor Angiogenesis in a Colon Cancer CT26 Tumor Model

Next, we investigated whether Erk1/2/HIF-1α/STAT3/VEGFA is critical for tumor angiogenesis under DOKD feeding. WB results showed that the signal transducer and activator of transcription 3 (STAT3), p-STAT3, and VEGFA expression were significantly decreased in the DOKD group compared with the ND group (Figure 8b–d; * *p* < 0.05, ** *p* < 0.01 and *** *p* < 0.001). These results suggested that DOKD inhibits tumor angiogenesis partly via inactivation of the Erk1/2/HIF-1*α*/STAT3/VEGFA signaling pathway.

### 3.9. HIF-1α Regulated Tumor Angiogenesis and Metastasis by Vimentin, MMP9 and VEGFA Pathway in CT26^+^ Colon Cancer Cells

To further investigate the effect of HIF-1*α* on the migration and angiogenesis of CT26 colon cancer cells, we inhibited the HIF-1*α* pathway using a HIF-1*α* inhibitor (LW6). Firstly, we examined whether LW6 could affect the viability of CT26^+^ cells using a CCK8 assay. Our results revealed the dose-dependent cytotoxicity of LW6 for CT26^+^ cells (Figure 9a). The concentration of LW6 was based on the manufacturer’s instructions. The results of CCK8 suggested that 400 μM LW6 be used for subsequent experiments. WB revealed that the expression of HIF-1*α*, MMP9, N-cadherin, VEGFA, and vimentin proteins was significantly decreased (Figure 9c–g; * *p* < 0.05, ** *p* < 0.01 and *** *p* < 0.001) in the LW6-treated group, suggesting that HIF-1*α* regulated tumor angiogenesis and metastasis by vimentin, MMP9, and VEGFA in CT26^+^ colon cancer cells.

## 4. Discussion

It has been confirmed that using ketone bodies as an energy supplying substance, such as in the KD, can effectively inhibit the development of many cancers [6,7,8,9]. Due to the absence of the inner mitochondrial membrane of tumor cells, the lack of enzymes capable of utilizing ketone bodies leads tumor cells to use glycolysis rather than oxidative phosphorylation for energy requirements; this characteristic of energy metabolism was summarized by Warburg in 1927 as the “Warburg effect” [49]. In addition, pyruvate is converted to lactic acid at a high rate of glycolysis, which forms an acidic tumor microenvironment and promotes tumor migration and invasion [50]. Therefore, exploiting the metabolic characteristics of cancer cells can provide new opportunities for therapeutic strategies.

To date, the KD’s anti-tumor mechanism is still remains unknown, and further optimization of the KD formulation is needed. With the deepening of the KD’s anti-tumor research, the KD’s selection of fatty acids has been optimized. Some defects in the existing KD formula may still lead to adverse reactions, such as increased cholesterol, poor palatability, vomiting, constipation, intestinal flora disorder, and so on [51]. However, the KD has the advantages of low cost, a great variety of available foods, and few side effects, so it can be used as a single therapy and in association with other treatments, which gives it a high research potential. Donkey oil contains essential minerals, vitamin E, and unsaturated fatty acids (oleic acid, linoleic acid, and palmitic acid); unsaturated fatty acids have important nutritional and functional properties. The KD is low in carbohydrates and high in fat, and previous studies have established the potential of KD as a supportive treatment against cancer. Thus, DO, with a high bioavailability and as a fat component of the KD, may enhance the anti-tumor effect of the KD. As we predicted, we found that DOKD inhibits tumor growth in CT26^+^-bearing mice. Moreover, the level of *β*-hydroxybutyrate was higher in the DOKD group than in the ND group. Compared with the ND group, there was no significant difference in blood glucose levels in the DOKD group, it suggested that DOKD-induced tumor growth inhibition could be mainly caused by ketone body *β*-hydroxybutyrate induction rather than blood glucose.

Inflammation pathways have emerged as promising targets for cancer therapy. Diakos et al. considered local inflammation to be the same as local immune response. A highly proliferative tumor grows rapidly and its blood supply is insufficient, leading to anoxic necrosis. Subsequent cell necrosis releases S100a9, which promotes immune cells to enter the tumor microenvironment [52]. Hu et al. established the colorectal cancer nude mouse model and concluded that S100a9 regulated inflammatory response and tumor progression by activating the TLR4/NF-*κ*B signaling pathway [45]. Moreover, S100a9 induces an immune response by activating natural killer cells [53]. In addition, Litak et al. suggested that the up-regulation of TLR4 expression can promote the expression of programmed death ligand-1 (PD-1L) and inhibit tumor development by enhancing the immune response [54]. Nowadays, there are not only studies on IL-17 promoting tumor, but also reports on its anti-tumor effects. IL-17 inhibits tumor growth and metastasis in colon cancer by enhancing the immune function of T cells and NK cells [55]. However, some evidence shows that IL-17 promotes tumor progression in various tumors, including melanoma, breast cancer, and liver cancer [56]. IL-17 induces inflammatory gene expression through the NF-*κ*B pathway as well as takes part in regulating tumor immunity [57]. In addition, the TLR4/MyD88/NF-*κ*B pathway has been reported to be relevant in the maturation and inflammatory response of immune cells. Notably, HIF-1*α* is vital for the activation of the IL-17 and TLR4/MyD88/NF-*κ*B pathways [58,59]. Here, we found that DOKD contributed to the up-regulation of S100a9, IL-17, NF-*κ*B p65, TLR4, MyD88, and TNF-*α* expression, and the down-regulation of HIF-1*α* expression, suggesting that DOKD may effectively reduce the growth of CT26^+^ tumor cells through the IL-17/HIF-1*α*/TLR4/NF-*κ*B p65 signaling pathway.

Tumor metastasis is one of the reasons for the high mortality of cancer patients [26]. Therefore, inhibiting tumor metastasis is critical for effective cancer therapy. HIF-1α plays a vital role in tumor metastasis [60,61]. N-cadherin, vimentin, and MMP9 are common molecules involved in metastasis. HIF-1*α* induces cell migration by increasing the gene expression of N-cadherin and vimentin and other mesenchymal markers. The results of Huang’s study indicate that the combination of dextran sulfate and the HIF-1*α* inhibitor inhibits gastric cancer development more effectively than each drug alone via inhibiting the expression of HIF-1*α* and N-cadherin [62]. Increased Src and Erk1/2 expression correlates with enhanced colon cancer metastasis [63,64]. Furthermore, Erk1/2 signaling can up-regulate the expression of transcription factors, including snail, and induce the expression of N-cadherin and fibronectin [65,66,67]. Moreover, MMP9 is highly expressed in various tumor tissues and promotes the progression of tumor cells [33]. Snail is reported to be a key tumor progression and metastasis regulator via increasing MMP9 expression and tumor invasion [68]. Li et al. demonstrated that HIF-1*α* and MMP9 promote tumor cell migration and invasion, and accelerate tumor progression [61]. Jia et al. demonstrated that downregulated HIF-1*α*/Snail/MMP9 proteins inhibit esophageal cancer cell invasion and metastasis [69]. Our data found that DOKD may suppress tumor development via adjusted metastasis and invasive through p-SRc/HIF-1*α*/Erk1/2/Snail and MMP9 signaling pathways. Meanwhile, the results of in vitro experiments further showed that LW6 (HIF-1*α* inhibitor) could effectively inhibit tumor metastasis by inhibiting the expressions of HIF-1*α*, MMP9, N-cadherin, and Vimentin.

In addition, the process of tumor angiogenesis is an important hallmark of cancer progression. Several lines of evidence indicate that angiogenic factors are involved in neoplastic growth and aggressiveness in tumors [34]. VEGFA was an effective stimulator of tumor angiogenesis [70]. It has been reported that HIF-1α stimulates angiogenesis by activating the expression of downstream target genes, including VEGFA and VEGFR [71]. The inhibition of HIF-1*α*/VEGFA can successfully suppress tumor growth, metastasis, and angiogenesis [72]. In addition, Erk signaling has been reported to be relevant in multiple tumor invasions [73] and to regulate the transcription factors for tumor angiogenesis [74]. A study demonstrated that the Erk1/2/HIF-1*α* signaling pathway might promote the angiogenesis of tumor cells by activating VEGFA [75]. Lin et al. reported that the Erk/HIF-1*α* signaling pathway might be related to the increased expression of VEGFA and that it promotes tumor angiogenesis [75]. STAT3 can affect tumor angiogenesis by regulating VEGF [76,77]. Moreover, STAT3 could regulate the activity and stability of HIF-1*α*. Morscher et al. observed that KD inhibited angiogenesis and tumor growth in vivo by suppressing VEGFA [78]. In the present study, we found that DOKD may exert its role in inhibiting CT26^+^ tumor angiogenesis by inhibiting the STAT3/HIF-1*α*/VEGFA pathway. Meanwhile, the results of in vitro experiments further showed that LW6 could effectively inhibit tumor angiogenesis by inhibiting the expressions of HIF-1α and VEGFA.

## 5. Conclusions

Our study revealed that DOKD therapy for CT26^+^-bearing mice resulted in the inhibition of inflammation, metastasis, and angiogenesis, as demonstrated by the IL-17/HIF-1*α*/TLR4/NF-*κ*B p65, HIF-1*α*/N-cadherin/Vimentin, and HIF-1*α*/STAT3/ VEGFA pathways. Therefore, DOKD might be a promising treatment for suppressing tumor development. These results preliminarily identified the antitumor mechanism of DOKD and laid a foundation for the production and application of DOKD. Although DOKD has a significant anti-tumor effect, dietary therapy is still an adjuvant anti-tumor therapy. How to optimize the formulation of DOKD and carry out combined therapy to achieve better anti-tumor outcomes needs further research.

## Figures and Tables

**Figure 1 genes-14-01024-f001:**
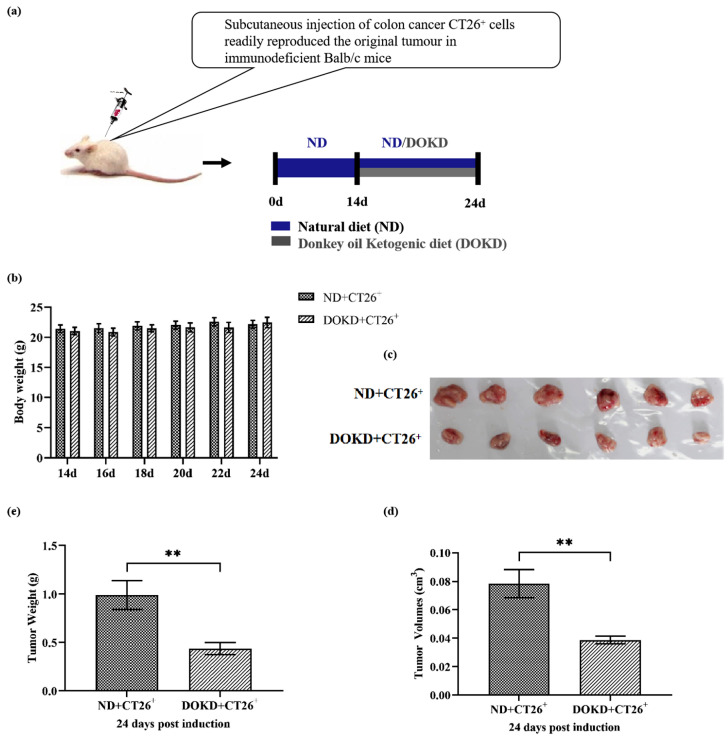
The effect of DOKD on body weights and tumors in mice. (**a**) Treatment strategy: after subcutaneous injection of CT26^+^ tumor cells, mice were given ND for 14 days, followed by ND or DOKD for 10 days. (**b**) The weights of mice were measured on days 14, 16,18, 20, 22 and 24. (**c**) The tumor morphology on day 24 after ND/DOKD treatment. (**d**) Tumor weight of each group on day 24 after ND/DOKD treatment. (**e**) The tumor volume of each group on day 24 after ND/DOKD treatment. Data are presented as mean ± SEM. Statistical significance: ** *p* < 0.01, n = 6 per group.

**Figure 2 genes-14-01024-f002:**
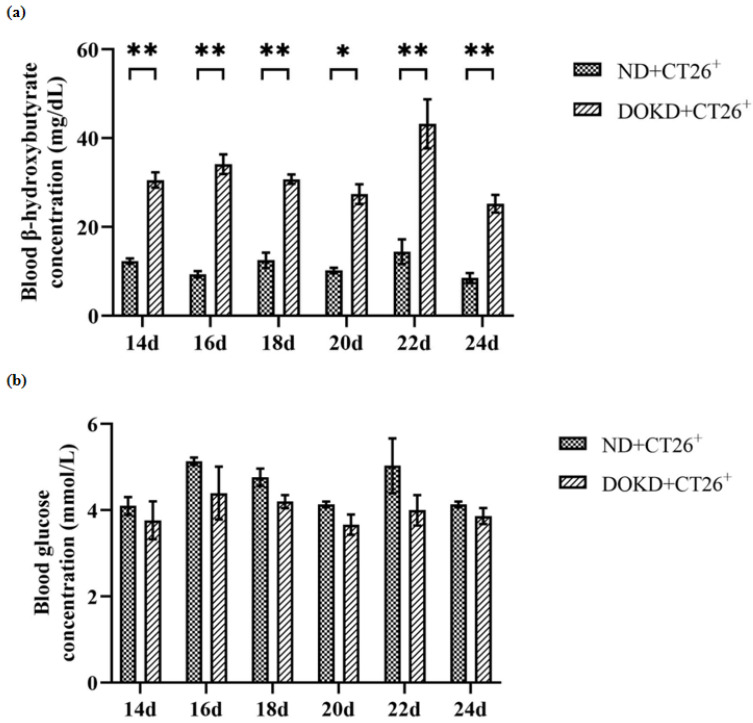
Effect of DOKD on blood *β*-hydroxybutyrate acid and blood glucose levels in CT26^+^ colon cancer mice. (**a**) The blood *β*-hydroxybutyric acid concentration was measured on days 14, 16, 18, 20, 22, and 24 with ND/DOKD treatment. The blood *β*-hydroxybutyrate acid level was measured at 8 am using a handheld ketone meter. (**b**) The blood glucose concentration was measured on days 14, 16, 18, 20, 22, and 24 with ND/DOKD treatment. The blood glucose level was measured at 8 am using a handheld ketone meter. Data are presented as mean ± SEM. Statistical significance: * *p* < 0.05 and ** *p* < 0.01, n = 6 per group.

**Figure 3 genes-14-01024-f003:**
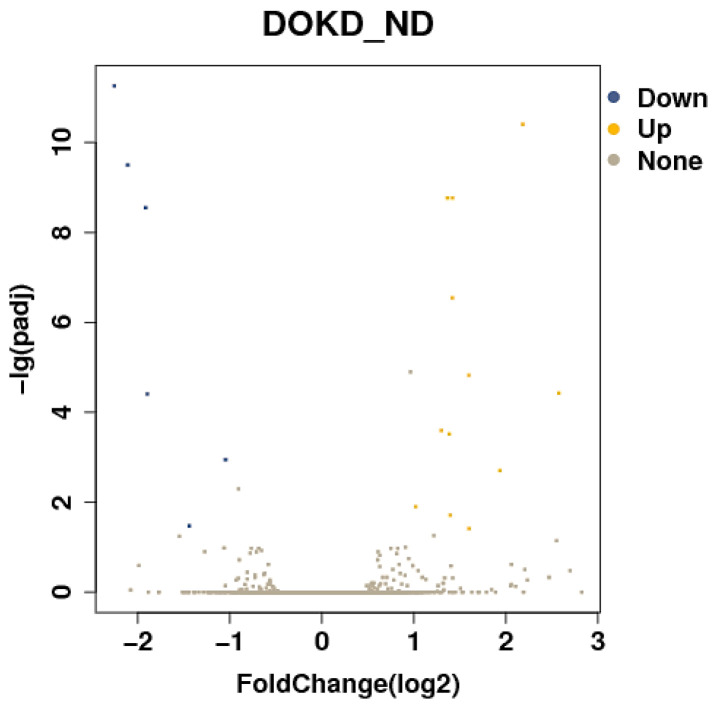
Volcano plot of differentially expressed genes.

**Figure 4 genes-14-01024-f004:**
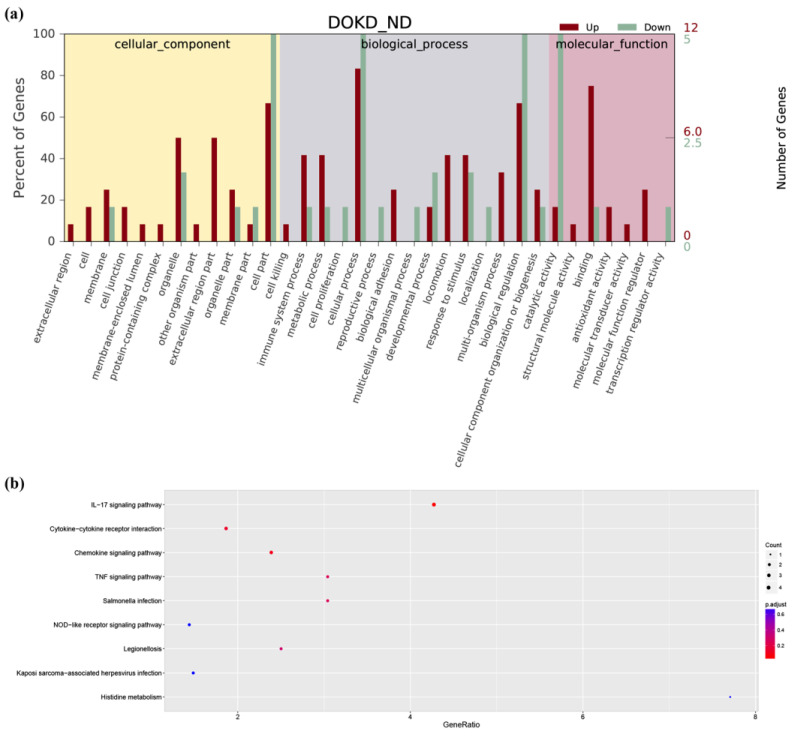
In ND/DOKD treated mice with CT26 tumors, the screened genes were enriched by GO, and the screened target genes were enriched by the top nine potential KEGG pathways. (**a**) The top 10 most significant GO term entries in each GO category were selected for display. (**b**) The top nine potential KEGG pathway enrichment of screened DEGs in DOKD treatment to CT26^+^ mice. The size of the points represents the number of DEGs in their representative pathways, and the corrected *p*-values were also highlighted with different colors.

**Figure 5 genes-14-01024-f005:**
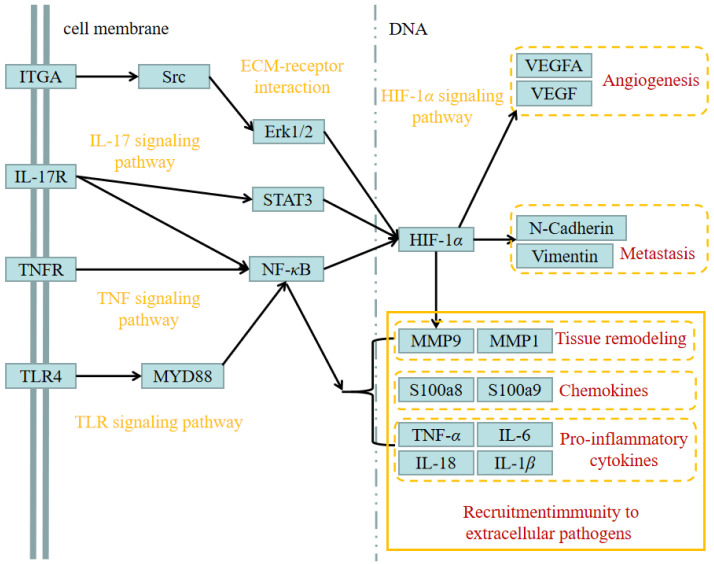
Systematic understanding of the underlying targets and treatment pathways of DOKD on mouse CT26 colon cancer. All the constructed treatment pathways were summarized by published articles.

**Figure 6 genes-14-01024-f006:**
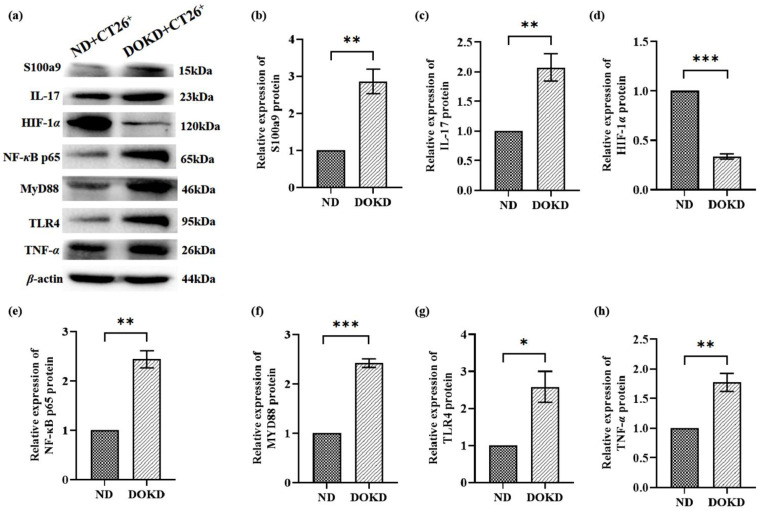
S100a9, IL-17, HIF-1*α*, NF-*κ*B p65, MYD88, TLR4, and TNF-*α* proteins were expressed in CT26^+^ mice with ND/DOKD treatment. (**a**) The representative expression of S100a9, IL-17, HIF-1*α*, NF-*κ*B p65, MYD88, TLR4, and TNF-*α* in CT26^+^ tumor tissues and *β*-actin as the internal control for standardization. Quantitative analysis of S100a9 (**b**), IL-17 (**c**), HIF-1*α* (**d**), NF-*κ*B p65 (**e**), MyD88 (**f**), TLR4 (**g**), TNF-*α* (**h**) proteins. Each experiment was repeated three times. All the data were presented as mean ± SEM. Independent-sample T-test was used to compare the data of different groups. Statistical significance: * *p* < 0.05, ** *p* < 0.01, *** *p* < 0.001, n = 3 per group.

**Figure 7 genes-14-01024-f007:**
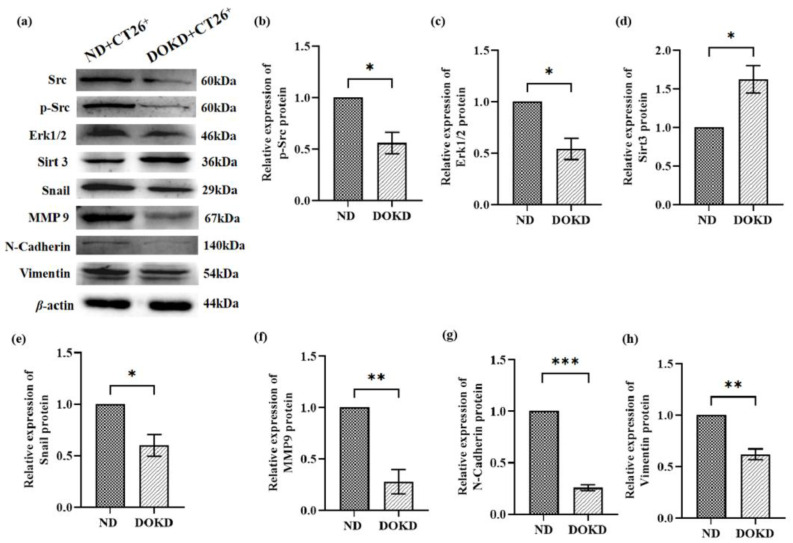
The expression of Src, p-SRc, Erk1/2, Sirt3, snail, MMP9, N-cadherin, and vimentin proteins in CT26^+^ bearing mice with ND/DOKD treatment. (**a**) Representative WB of Src, p-SRc, Erk1/2, Sirt3, snail, MMP9, N-cadherin, and Vimentin expression in CT26^+^ tumor and *β*-actin as the internal control for standardization. Quantitative analysis of p-Src (**b**), Erk1/2 (**c**), Sirt3 (**d**), snail (**e**), MMP9 (**f**), N-cadherin (**g**), and vimentin (**h**) proteins in ND/DOKD group. Each experiment was repeated three times. All the data were presented as mean ± SEM. Independent-sample T-test was used to compared the data of different groups. Statistical significance: * *p* < 0.05, ** *p* < 0.01, *** *p* < 0.001, n = 3 per group.

**Figure 8 genes-14-01024-f008:**
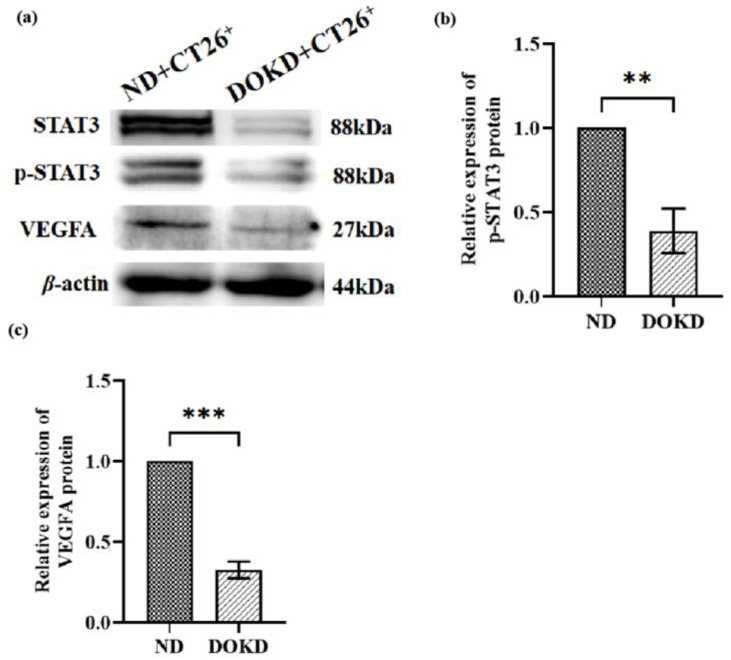
STAT3, p-STAT3, and VEGFA protein expression in CT26^+^ colon cancer mouse model with ND/DOKD treatment. (**a**) Representative WB of STAT3, p-STAT3, and VEGFA expressions in CT26^+^ tumor tissues and *β*-actin as the internal control for standardization. Quantitative analysis of p-STAT3 (**b**), VEGFA (**c**) proteins. Each experiment was repeated three times. All the data were presented as mean ± SEM. Independent-sample T-test was used to compare the data of different groups. Statistical significance: ** *p* < 0.01, *** *p* < 0.001, n = 3 per group.

**Figure 9 genes-14-01024-f009:**
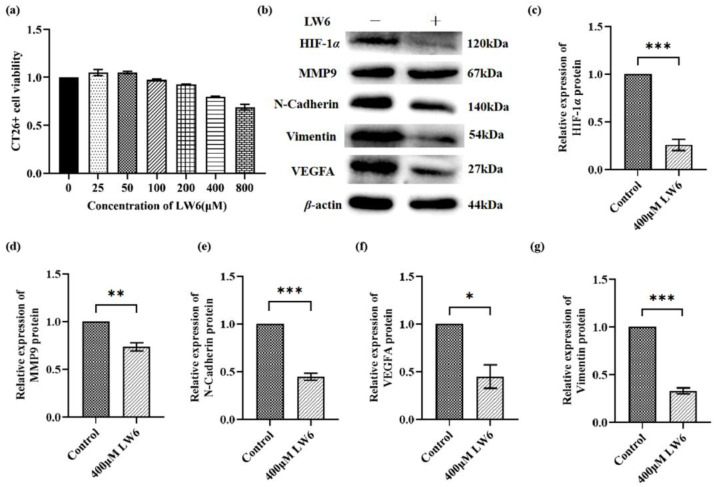
HIF-1*α* regulated the expression of N-cadherin, vimentin, VEGFA, and MMP9 in CT26^+^ colon cancer cells. The viability of CT26^+^ cells was detected by CCK8 with LW6 treatment (**a**). (**b**) Representative WB of HIF-1*α*, MMP9, N-cadherin, VEGFA, and vimentin expressions in CT26^+^ cells treated with/without LW6, with *β*-actin as the internal control for normalization. Quantitative analysis of HIF-1*α* (**c**), MMP9 (**d**), N-cadherin (**e**), VEGFA (**f**), and vimentin (**g**) proteins in LW6-treated groups. Each experiment was repeated three times. All the data were presented as mean ± SEM. Independent-sample T-test was used to compare the data of different groups. Significant differences between groups are indicated by an asterisk (* *p* < 0.05, ** *p* < 0.01, *** *p* < 0.001), n = 3 per group.

**Table 1 genes-14-01024-t001:** Detailed List of Macronutrient Components of ND and DOKD.

Composition	ND		DOKD	
	Weight (grams/kg)	Energy Density (kcal/g)	Weight (grams/kg)	Energy Density (kcal/g)
Casein	200.000	0.800	162.500	0.650
Carbohydrate	626.000	2.504	-	-
-Corn Starch	394.000	1.576	-	-
-Maltodextrin	132.000	0.528	-	-
-Sucrose	100.000	0.400	-	-
Fat	70.000	0.630	690.000	6.210
-Soybean oil	70.000	0.630	-	-
-Donkey oil	-	-	690.000	6.210
Cellulose	53.500	0.000	97.000	0.000
L-cysteine	3.000	0.012	3.000	0.012
Mineral mixture	35.000	0.000	35.000	0.000
^a^ Fiber mixture	10.000	0.040	10.000	0.040
Choline tartrate	2.500	0.000	2.500	0.000
Antioxidants (TBGQ)	0.014	0.000	0.014	0.000
Total (g)	1000.014	-	1000.014	-
The energy density (kcal/g)		3.9		6.912

ND: Natural diet. DOKD: Donkey oil-based ketogenic diet. The standard diet was based on the composition of AIN-93G. ^a^ Fiber mixture (V1002): containing 99.4% (*w*/*w*) starch.

## Data Availability

All data generated or analyzed during this study are included in this article.

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
