# Peer review of "Donkey Oil-Based Ketogenic Diet Prevents Tumor Progression by Regulating Intratumor Inflammation, Metastasis and Angiogenesis in CT26 Tumor-Bearing Mice"

_genes, 2023, doi:10.3390/genes14051024_

Round 1

Reviewer 1 Report

The present article evaluated the implications of the donkey oil based ketogenic diet in preventing tumor progression by regulating inflammatory, apoptotic, and metastatic metabolic pathways. The topic is relevant, but there is a need for major changes to improve the initial form:

Shape suggestions

Abbreviations are explained when they first appear in the abstract or main text and contribute to making the text easier to read and the information conveyed more efficiently. Once an abbreviation has been established and explained, it will be used throughout the entire manuscript, with the exception of the abstract, where it must be treated separately. Please revise the whole manuscript and explain the abbreviations used directly, without explanation (e.g., abstract L22-23, the ones that are used more than once should be explained at the first mention, the other ones should be written directly in the long form).

It is not necessary to bold certain words within a table, especially if they are not at the head of the table.

Abbreviations in tables should be explained in the legend below the table.

Subsection 2.4./3.6. – there is no need for a punctuation mark in the title.

Bibliographic reference number 13 must be used in English.

Content suggestions

Treatment strategies in colorectal cancer need to be developed to better understand what the unmet needs are and to what extent they can be addressed by ongoing research.

As the ketogenic diet is a focus of this research, it is important to develop the molecular mechanisms with implications on metabolic changes based on this diet. I suggest checking and referring to: PMID: 33121986.

The aim of the paper presented in the last paragraph of the introduction needs to be improved from the perspective of describing the contribution to the field under analysis and the elements of scientific novelty presented.

The results section should not contain bibliographical references because only the authors' own results are presented. Everything containing references is organized in the discussion section, where comparisons can be made with other studies, supplementary explanations, etc. Please revise the results section from this point of view.

It is advisable to present the potential side effects of the ketogenic diet and the measures necessary to improve their management. I suggest checking and referring to: PMID: 32820459.

As the last paragraph of the Discussion section, it is advisable to detail the strengths, but especially the limitations, of your study and to what extent they could be resolved in view of future research directions, especially since the authors didn’t provide a separate conclusion section. 

Author Response

Dear Reviewer:

First of all, we would like to express our great appreciation to you. We have fully addressed the concerns raised by the reviewers in the reports, and the manuscript has been revised accordingly. A point by point response to the all comments has been supplied. All comments in blue italic type, our response in boldface type. We thank the referees for careful reading, and constructive suggestions that enable us to improve our manuscript.

Reviewer 2 Report

In the manuscript entitled "Donkey Oil Based Ketogenic Diet Prevents Tumor Progression by Regulating Intratumor Inflammation, Metastasis and Angiogenesis in Colon 26 Tumor-Bearing Mice", the authors demonstrated that DOKD therapy in CT26+ bearing mice inhibits inflammation, metastasis, and angiogenesis, as demonstrated by IL-17/HIF-1α/TLR4/NF-κB p65, HIF-1α/N-cadherin/Vimentin and HIF-1α/STAT3/VEGFA pathways. Some suggestions are list below.

1.      RNA-seq was used to detect DEGs between DOKD and ND. Are the authors sure only 18 DEGs exist? If so, PLS draw the Volcanoplot of all genes (significant or not significant). And I do not understand how the authors did the GO enrichment analysis and KEGG pathway analysis by just analyzing these 18 DEGs.

2.      RNA-seq=RNA sequencing. In line 139, RNA-sequencing sequencing?

3.      WB results: “The density of the band was presented in scanning units, and the expression levels were quantified by comparison with the levels of the control.” Then the value of control group should be one.

4.      The conclusions that “DOKD reduces the metastasis and invasive abilities of CT26+ colon cancer cells partly via SRc/HIF-1α/Erk1/2/Snail /MMP9 signaling pathway in vivo” or “DOKD inhibits tumor angiogenesis partly via inactivation of the Erk1/2/HIF-1α/STAT3/VEGFA signaling pathway.” are not appropriate by just detecting the expression levels of these proteins.

Author Response

(The authors gave the same response as above.)

Round 2

Reviewer 1 Report

No correction is highlighted on the manuscript to be checked (use different colour or diferent coloured background).

A lot of things are mentioned that have been done in the letter of the authors, but they cannot be found in the manuscript (i.e. check that all abbreviations mentioned in the figures/tables to be explained under them!). Please check my previous report and respond VISIBLE in the manuscript to my suggestions. In all revisions you will do in the future, highlight in a visible way the changes/completions.

L39-40. Develop better and complete the idea regarding the effects of different types of treatments and their monitoring in colon cancer - I suggest checking and referring to Pallag, A.; et al. Monitoring the effects of treatment in colon cancer cells using immunohistochemical and histoenzymatic techniques. Rom. J. Morphol. Embriol., 56(3), 2015, 1103-1109. PMID: 26662146

I am not seeing the novelty of the study in the last paragraph of Introduction. Please highlight it. Nor the strengths and limitations of your research in the last paragraph of Discussion (after making the actual last paragraph the future Conclusions section).

2. Materials and methods section

-        the Model, Producer/manufacturer, City and Country for EACH APPARATUS used in the research, and 

-        the Producer, Country, purity degree, and concentration used for each REAGENT/chemical used.  

- Subsection 2.2. How many mice have been used in the experimental part?

Subsection 2.9. The softs used for data analysis must be referenced. Please see https://www.ibm.com/support/pages/how-cite-ibm-spss-statistics-or-earlier-versions-spss

Conclusions section is missing. Made the last paragraph a separate section, of course - removing "In conclusion," as it would be duplicated and useless.

Author Response

(The authors gave the same response as above.)
